# Comparing perceived clarity of information on overdiagnosis used for breast and prostate cancer screening in England: an experimental survey

Alex Ghanouni, Cristina Renzi, Emily McBride, Jo Waller

► Prepublication history and additional material are available. To view these files please visit the journal online (http://dx.doi.org/10.1136/bmjopen-2017-015955).

Research Department of Behavioural Science and Health, University College London, London, UK

**Correspondence to**
Dr Jo Waller; j.waller@ucl.ac.uk

## ABSTRACT

**Objectives** 'Overdiagnosis', detection of disease that would never have caused symptoms or death, is a public health concern due to possible psychological and physical harm but little is known about how best to explain it. This study evaluated public perceptions of widely used information on the concept to identify scope for improving communication methods.

**Design** Experimental survey carried out by a market research company via face-to-face computer-assisted interviews.

**Setting** Interviews took place in participants' homes.

**Participants** 2111 members of the general public in England aged 18–70 years began the survey; 1616 were eligible for analysis. National representativeness was sought via demographic quota sampling.

**Interventions** Participants were allocated at random to receive a brief description of overdiagnosis derived from written information used by either the NHS Breast Screening Programme or the prostate cancer screening equivalent.

**Primary and secondary outcome measures** The primary outcome was how clear the information was perceived to be (extremely/very clear vs less clear). Other measures included previous exposure to screening information, decision-making styles and demographic characteristics (eg, education). Binary logistic regression was used to assess predictors of perceived clarity.

**Results** Overdiagnosis information from the BSP was more likely to be rated as more clear compared with the prostate screening equivalent (adjusted OR: 1.43, 95% CI 1.17 to 1.75; p=0.001). Participants were more likely to perceive the information as more clear if they had previously encountered similar information (OR: 1.77, 1.40 to 2.23; p<0.0005) or a screening leaflet (OR: 1.35, 1.04 to 1.74; p=0.024) or had a more 'rational' decision-making style (OR: 1.06, 1.02 to 1.11; p=0.009).

**Conclusions** Overdiagnosis information from breast screening may be a useful template for communicating the concept more generally (eg, via organised campaigns). However, this information may be less well-suited to individuals who are less inclined to consider risks and benefits during decision-making.

## INTRODUCTION

Overdiagnosis, defined as the detection of disease that would never have caused

### Strengths and limitations of this study

► There has been very little research regarding how to effectively communicate the concept of overdiagnosis to lay people, which this study aims to begin to address.
► It benefited from measuring a wide range of predictor variables in a large sample of the general public in which quotas were set with the aim of achieving population-representativeness.
► Information on overdiagnosis was extracted from information sources and presented separately, meaning that perceived clarity may be different when read within the original contexts.
► Participants' objective understanding of the information was not measured and it is unknown how well this correlates with perceived clarity.
► The sample was also heterogeneous, meaning that findings may not apply equally to a specific healthcare context in which overdiagnosis may be an issue.

symptoms or death within a person's lifetime, is a topic of widespread concern in healthcare[1] since it results in negative psychological consequences, overtreatment and overutilisation of scarce resources.[2] This has led to various efforts to communicate the issue to patients and the public, increasing awareness so that it can be factored into decision-making around health.[1 3–7]

However, these efforts have proved challenging. For example, there is strong, consistent evidence that public awareness of overdiagnosis remains markedly low, both in the UK and internationally.[8–10] Fewer than one in three individuals surveyed in the UK reported recognising the term and virtually no participants were able to define it accurately.[11] This demonstrates that there is considerable scope for improving understanding of the concept, although research into effective communication methods is limited.[2] The available studies have also

shown that the concept is difficult to explain in a way that lay people find clear. The term is often considered counterintuitive[12 13] since it contradicts an established faith in the importance of early detection.[14] A perceived lack of clarity of information on overdiagnosis is problematic since it is likely to result in feelings of uncertainty, contributing to decisional conflict.[15]

Breast and prostate cancer screening are two notable contexts in which overdiagnosis is a prominent issue.[16] In the UK, a thorough review of written information materials was carried out for the Breast Screening Programme with the explicit aim of improving communication of the concept.[17 18] Similarly, men receiving care under the Prostate Cancer Risk Management Programme are informed about overdiagnosis, primarily by clinicians and also via associated leaflets.[19 20] These written materials offer two brief descriptions of the concept that could potentially serve as models for communicating it more generally (eg, to screening invitees or the wider public).

Strategies for communicating overdiagnosis may also become apparent via a better understanding of factors associated with perceived clarity of explanations, such as people's approaches to making important decisions ('decision-making styles' (DMS)).[21] We previously explored whether this was associated with the amount of cancer screening information leaflets that screening-eligible individuals had read; less 'rational' participants who tended not to weigh up available information were less likely to have read information in leaflets used by bowel and cervical screening programmes.[22] Since existing written information has been designed to allow individuals to consider risks and benefits, it is plausible that people who are less inclined to make decisions in this way would find materials less clear. There may also be relevant associations with other DMS (eg, 'avoidant' or 'dependent' styles) or demographic characteristics (eg, education and previous exposure to screening information).

We carried out a large-scale experimental survey of the general public in England to assess perceived clarity of existing, widely-used information on overdiagnosis derived from written material for breast and prostate cancer screening. We compared perceived clarity of these two pieces of information and tested for associated characteristics (specifically, DMS, demographics and previous exposure to screening information).

## METHOD
### Design
Institutional ethical approval was obtained from the University College London Research Ethics Committee (5771/002). The study design and some aspects of the measures have also been described elsewhere.[22] Data were collected between April and May 2016, as part of Wave 3 of the Attitudes, Behaviour and Cancer UK Survey (ABACUS). A market research agency (TNS) conducted face-to-face computer-assisted interviews in people's homes, as part of their weekly omnibus survey in England. Participants were presented with information describing overdiagnosis from either the leaflet used by the NHS Breast Screening Programme or the equivalent used by the Prostate Cancer Risk Management Programme in England, allocated at random in a 1:1 ratio. Participants were told:

> 'We would now like to ask you some questions about leaflets on health-related topics. The NHS offers people a variety of screening tests to check for illnesses before symptoms have appeared. People offered an NHS screening test are often given a leaflet that explains the risks and benefits of having the test. The leaflet is either posted or given out by a doctor or nurse.'

After completing items on previous exposure to screening information (described below), participants were given one of the following pieces of information:

> 'The test can find an illness that would never have caused a person harm. Some people will be diagnosed and treated for an illness that would never otherwise have been found and would not have become life-threatening.' (From the breast screening information leaflet)[18]

> 'The test may make you worry by finding an illness that may never cause any symptoms or shorten your life.' (From the prostate screening information sheet)[20]

The provided descriptions consisted of all the text relating to overdiagnosis in the prostate screening leaflet (as it existed at the time the survey was designed) and all relevant conceptual information from the breast screening leaflet (ie, omitting overlapping but context-specific information on ductal carcinoma in situ and ratios of breast cancer deaths prevented to overdiagnosed cases). Text that reiterated the information above with slightly different wording was also omitted. The breast screening leaflet is freely available online; the original prostate screening leaflet is available on request.

### Participants
Random location sampling was conducted using Census statistics and the Postcode Address File. National representativeness was achieved via quotas for demographic characteristics (eg, employment status and gender). Participants were eligible if they were aged between 18 and 70 years. The total number of participants approached for this study was determined by funding constraints for the broader ABACUS survey.

### Measures
Demographics: Principal measures were gender, ethnicity, marital status, highest level of education qualification obtained and age. We also measured social class grade (a widely used classification of socioeconomic status based on type of employment).[23] Participants also stated whether they had been diagnosed with cancer (and what type, if applicable) or if they knew anyone who had been diagnosed.

Previous exposure to screening information: Participants were asked three questions:

*'Have you ever read a leaflet from the NHS about a screening test?'*

*'Have you ever read information about a screening test on an NHS website?'*

*'Have you ever talked with a doctor or nurse about a screening test?'*

(Response options: *'Yes'*, *'no'*, *'not sure'*).

If eligible for either cervical, breast or bowel screening programmes in England and anticipated not to find the question distressing, participants were asked to state their previous screening history using up to four items based on the Precaution Adoption Process Model (PAPM),[24] for example,

*'The next set of questions are about breast screening. The NHS breast screening programme invites women to have regular mammograms (x-rays of their breasts). 'Which one of the following best describes you?'*

Response options were:

*'I have never heard of breast screening, I have heard of breast screening but have never been invited, I have been invited to breast screening but have never been, I have been invited to breast screening but have not been every time I was invited, I have been invited to breast screening and have been every time I was invited'*

Since prostate screening in England is carried out via direct contact with clinicians rather than an organised programme, previous participation was assessed in the following way among eligible participants:

*'Men can have a blood test, called a prostate-specific antigen (PSA) test, to look for a protein in their blood that can be an indicator of prostate cancer. Which of the following best describes you?'*

Response options were:

*'I have never heard of a PSA test, I have heard of a PSA test but have never discussed it with a doctor or nurse, I have discussed a PSA test with a doctor or nurse but have never had it, I have discussed a PSA test with a doctor or nurse and have had the test, I have had the PSA test but I have never discussed it with a doctor or nurse.'*

Participants were not asked these questions if they had previously been diagnosed with the applicable type of cancer or if they were ineligible in terms of their gender or age (ie, not aged 18–64 years for cervical screening questions, 47–70 years for breast screening questions, 58–70 for bowel screening questions or 45–70 years for prostate screening questions).

Decision-making styles: Participants were informed:

*'We would like to start by asking you some questions about how you make important decisions. I am going to read out some statements describing how individuals go about making important decisions and I would like you to tell me how much you agree or disagree with each one.'*

Tendencies towards using each decision-making style were measured using 25 previously developed items that have been reported to have content, concurrent and construct validity.[21] These relate to five subscales, measured with five items each: 'Avoidant' (eg, *I postpone decision making whenever possible*), 'dependent' (eg, *I rarely make important decisions without consulting other people*), 'intuitive' (eg, *I generally make decisions that feel right to me*), 'rational' (eg, *My decision-making requires careful thought*) and 'spontaneous' (eg, *I make quick decisions*). Item order was randomised for each participant. Reliability was good for all five subscales within the analysed sample (Cronbach's α: 0.71–0.81).

Overdiagnosis information: Text on overdiagnosis was followed by the question, *'How clear do you find this description of a risk of the test?'*. Response options were:

*'Extremely clear'*, *'very clear'*, *'moderately clear'*, *'slightly clear'*, and *'not at all clear'*. Participants were also asked *'Have you ever read or heard similar information about a screening test?'*

(Response options: *'Yes'*, *'no'*, *'not sure'*).

### Piloting

Piloting aimed to ensure that the survey was acceptable and manageable for participants and that items were comprehensible. This consisted of a series of cognitive interviews[25] in which the survey was administered to 11 members of the general public via telephone, who were asked for feedback on whether any items were unclear, difficult to understand or offensive, followed by a 'soft-launch' of the survey to 431 participants to ensure data were useable.

### Analysis

Participants were excluded if data were missing, if they responded *'not sure'* or *'don't know'*, to any item or had a non-ordinal level of education.

Items relating to each decision-making style were scored from 1 to 5 and summed to create an overall score (higher scores represented a greater tendency to use a given approach to decision-making) and highest level of education was categorised based on Levels 1–4 used by the Office of National Statistics.[26] Social class grade was categorised as *'Grades A or B'*, *'Grades C1 or C2'* or *'Grades D or E'*. Ethnicity and marital status were dichotomised into *'White British'* versus *'Other ethnic groups'* and *'Married or living as a couple'* versus *'Single, widowed, divorced or separated'*, respectively.

Descriptive statistics were used to summarise the analysed sample overall and within each response category of perceived clarity of the overdiagnosis information. Responses on PAPM items were also summarised (previous screening participation vs no previous participation) separately for the four screening modalities for all individuals who completed them (ie, excluding those

who were not eligible to be asked the question and those who declined to answer).

The main analysis tested the null hypothesis that overdiagnosis text (from breast vs prostate information), demographics characteristics, previous exposure to screening information and scores on the five DMS scales were not associated with the primary outcome (perceived clarity of information). The assumption of proportional odds was violated (test of parallel lines: p=0.005), indicating that associations between independent variables and the outcome were not consistent for all possible ways of dichotomising perceived clarity (eg, *'extremely'* or *'very clear'* vs *'moderately'*, *'slightly'* or *'not at all clear'*). Hence, binary logistic regression was used instead of an ordinal logistic model, in which the outcome was dichotomised using the previously stated thresholds, since the authors considered this most conceptually meaningful. The authors made a collective judgement regarding the level of perceived clarity that should be considered appropriate but other thresholds can also be used. Tables 1 and 2 illustrate the distribution of responses in all levels and online supplementary appendixes 1 and 2 show the results of a sensitivity analysis in which the outcome is dichotomised at all other possible thresholds.

Demographic predictor variables used were gender, marital status, ethnicity, highest level of education, social class grade, previous diagnosis of cancer and knowing someone diagnosed with cancer. Variables related to previous exposure to screening information consisted of having previously (1) read a leaflet about screening, (2) read an NHS website about screening, (3) discussed screening with a doctor or nurse and (4) read or heard similar information. Scores on the five DMS subscales were also included. The remaining predictor variable was text on overdiagnosis. There was minimal (multi)collinearity between predictor variables, meaning that associations between predictors and the outcome were unlikely to be notably affected by the inclusion or omission of other predictors (variance inflation factors were ≤1.830). There were no violations of the assumption of linearity for continuous independent variables (Box-Tidwell procedure: all p values>0.114), indicating that associations with the outcome were consistent across all values.

Adjusted ORs for rating the information as 'extremely' or 'very clear' versus reporting a lower category of perceived clarity are reported with 95% CIs and p values, alongside descriptive statistics.

## RESULTS
### Participant characteristics
Interviews were carried out with 2111 participants; 495 were excluded due to missing or inapplicable data (see online supplementary appendix 3), leaving a total of 1616 in the main analysis. Percentages of participants who reported previously participating in screening were 75.2%, 80.5%, 60.9% and 16.6% for cervical, breast, bowel and prostate screening, respectively (n=732;

n=343; n=407; n=349). Other sample characteristics are described in table 1 and table 2.

### Predictors of rating the information on overdiagnosis as extremely/very clear
Participants were more likely to rate the breast overdiagnosis information as extremely or very clear (vs moderately, slightly or not at all clear) compared with the relevant information for prostate screening (OR: 1.43; 95% CI 1.17 to 1.75; p=0.001). Previous exposure to screening information was also associated: participants who had read or heard similar information before (1.77, 1.40 to 2.23; p<0.0005) or had read a screening leaflet (1.35, 1.04 to 1.74; p=0.024) were more likely to rate the information as extremely/very clear. However, previously reading an NHS website about screening (0.97, 0.72 to 1.30; p=0.815) and talking to a healthcare professional about screening (1.08, 0.83 to 1.39; p=0.576) was not associated (table 1).

Rational decision-making style scores were associated: participants with higher scores were more likely to rate the information as extremely/very clear (1.06, 1.02 to 1.11; p=0.009). There was moderate evidence against the null hypothesis for a relationship in the opposite direction in the case of the dependent decision-making style (0.97, 0.97 to 1.00; p=0.052). Trends are illustrated in figure 1. There were no other associations between other variables and the outcome (tables 1 and 2).

## DISCUSSION
In this survey of the general public in England, less than half of the sample rated either description of overdiagnosis as either extremely or very clear (38.4% and 46.8% for prostate and breast screening respectively). This is consistent with previous research reporting that lay people find the concept confusing and counterintuitive.[12 13] It also indicates that both pieces of information have room for improvement. To date, there has been little experimental research exploring the most effective methods of communicating overdiagnosis and related concepts in a way that would improve perceived clarity and potentially ameliorate decisional conflict;[15] previous research has assessed effects of manipulating disease terminology on treatment preferences (eg, 'abnormal cells' as an alternative to 'ductal carcinoma in situ' in the case of breast cancer)[27] and effects of information formats on comprehension of overdiagnosis information in screening (text, fact boxes and visual aids).[28] To our knowledge, this is the first study to compare how clear the public perceived two existing pieces of widely used information on overdiagnosis to be.

The information from the breast screening leaflet was somewhat more likely to be rated as clear than the equivalent information for prostate screening. This may be attributable to the intensive multistage development process that underpinned the revised breast information leaflet, consisting of a 'citizens' jury', expert input, cognitive testing with prospective service users and combining feedback from experts and lay people. The main practical implication of this finding

**Table 1** Perceived clarity of overdiagnosis information: descriptive statistics for categorical/ordinal variables, adjusted ORs, 95% CIs, p values for variables in the multivariable binary logistic regression model

| Characteristic | Total | 'How clear do you find this description of a risk of the test?' n (%) | | | | | Adjusted OR, 95% CI; p value |
| --- | --- | --- | --- | --- | --- | --- | --- |
| | | Extremely clear | Very clear | Moderately clear | Slightly clear | Not at all clear | Extremely/very clear (vs Not/slightly/moderately) |
| | (n=1616) | (n=148; 9.2%) | (n=542; 33.5%) | (n=592; 36.6%) | (n=173; 10.7%) | (n=161; 10.0%) | |
| Overdiagnosis information | | | | | | | |
| Breast screening text | 850 | 75 (8.8) | 321 (37.8) | 285 (33.5) | 81 (9.5) | 88 (10.4) | 1.43, 1.17 to 1.75; **0.001** |
| versus Prostate screening text | 766 | 73 (9.5) | 221 (28.9) | 307 (40.1) | 92 (12.0) | 73 (9.5) | |
| Gender | | | | | | | |
| Male | 748 | 66 (8.8) | 228 (30.5) | 303 (40.5) | 81 (10.8) | 70 (9.4) | 0.92, 0.74 to 1.14; 0.447 |
| versus Female | 868 | 82 (9.4) | 314 (36.2) | 289 (33.3) | 92 (10.6) | 91 (10.5) | |
| Ethnicity | | | | | | | |
| White British | 1229 | 116 (9.4) | 425 (34.6) | 447 (36.4) | 121 (9.8) | 120 (9.8) | 1.19, 0.92 to 1.54; 0.197 |
| versus Other ethnic groups | 387 | 32 (8.3) | 117 (30.2) | 145 (37.5) | 52 (13.4) | 41 (10.6) | |
| Marital status | | | | | | | |
| Married or living as a couple | 987 | 92 (9.3) | 344 (34.9) | 355 (36) | 101 (10.2) | 95 (9.6) | 1.16, 0.93 to 1.45; 0.184 |
| versus single, widowed, divorced or separated | 629 | 56 (8.9) | 198 (31.5) | 237 (37.7) | 72 (11.4) | 66 (10.5) | |
| Highest level of education | | | | | | | Overall p-value: 0.482 |
| No formal qualifications | 275 | 24 (8.7) | 97 (35.3) | 107 (38.9) | 25 (9.1) | 22 (8.0) | 1.24, 0.86 to 1.79; 0.248 |
| Approximately Level 1, 2 or 3 | 858 | 68 (7.9) | 297 (34.6) | 312 (36.4) | 97 (11.3) | 84 (9.8) | 1.13, 0.87 to 1.46; 0.353 |
| versus Approximately Level 4 | 483 | 56 (11.6) | 148 (30.6) | 173 (35.8) | 51 (10.6) | 55 (11.4) | |
| Social class grade | | | | | | | Overall p value: 0.610 |
| Grade A or B | 351 | 41 (11.7) | 117 (33.3) | 122 (34.8) | 31 (8.8) | 40 (11.4) | 1.18, 0.85 to 1.64; 0.323 |
| Grade C1 or C2 | 786 | 71 (9.0) | 267 (34.0) | 298 (37.9) | 80 (10.2) | 70 (8.9) | 1.09, 0.85 to 1.40; 0.507 |
| versus Grade D or E | 479 | 36 (7.5) | 158 (33.0) | 172 (35.9) | 62 (12.9) | 51 (10.6) | |
| Personal diagnosis of cancer | | | | | | | |
| Yes | 86 | 11 (12.8) | 36 (41.9) | 25 (29.1) | 6 (7.0) | 8 (9.3) | 1.34, 0.84 to 2.12; 0.218 |
| versus No | 1530 | 137 (9.0) | 506 (33.1) | 567 (37.1) | 167 (10.9) | 153 (10.0) | |
| Knows someone with cancer | | | | | | | |
| Yes | 946 | 89 (9.4) | 348 (36.8) | 313 (33.1) | 97 (10.3) | 99 (10.5) | 1.23, 0.99 to 1.53; 0.060 |
| versus No | 670 | 59 (8.8) | 194 (29.0) | 279 (41.6) | 76 (11.3) | 62 (9.3) | |
| Previously read a screening leaflet | | | | | | | |
| Yes | 907 | 101 (11.1) | 340 (37.5) | 298 (32.9) | 87 (9.6) | 81 (8.9) | 1.35, 1.04 to 1.74; **0.024** |
| versus No | 709 | 47 (6.6) | 202 (28.5) | 294 (41.5) | 86 (12.1) | 80 (11.3) | |
| Previously read an NHS screening website | | | | | | | |
| Yes | 275 | 33 (12.0) | 105 (38.2) | 88 (32.0) | 23 (8.4) | 26 (9.5) | 0.97, 0.72 to 1.30; 0.815 |

Continued

**Table 1** Continued

| Characteristic | Total | Extremely clear | 'How clear do you find this description of a risk of the test?' n (%) | | | | Adjusted OR, 95% CI; p value |
| | | | Very clear | Moderately clear | Slightly clear | Not at all clear | Extremely/very clear (vs Not/slightly/moderately) |
| --- | --- | --- | --- | --- | --- | --- | --- |
| | (n=1616) | (n=148; 9.2%) | (n=542; 33.5%) | (n=592; 36.6%) | (n=173; 10.7%) | (n=161; 10.0%) | |
| versus No | 1341 | 115 (8.6) | 437 (32.6) | 504 (37.6) | 150 (11.2) | 135 (10.1) | |
| Discussed screening with doctor/nurse | | | | | | | |
| Yes | 535 | 67 (12.5) | 200 (37.4) | 172 (32.1) | 46 (8.6) | 50 (9.3) | 1.08, 0.83 to 1.39; 0.576 |
| versus No | 1081 | 81 (7.5) | 342 (31.6) | 420 (38.9) | 127 (11.7) | 111 (10.3) | |
| Previously read or heard similar information | | | | | | | |
| Yes | 584 | 78 (13.4) | 239 (40.9) | 189 (32.4) | 49 (8.4) | 29 (5.0) | 1.77, 1.40 to 2.23; **<0.0005** |
| versus No | 1032 | 70 (6.8) | 303 (29.4) | 403 (39.1) | 124 (12.0) | 132 (12.8) | |

Adjusted ORs and 95% CIs are relative to a stated reference category; p values<0.05 are in bold; all predictor variables are included in the model.

is that despite a (large) minority of participants rating the breast screening information as less than very clear, it may be more useful to adapt overdiagnosis information from this source when attempting to describe the concept in other contexts (rather than using the original information from prostate screening). For example, the NHS Abdominal Aortic Aneurysm Programme also describes overdiagnosis with the following text: '… around 54 out of every 10 000 men screened will eventually have surgery to repair an aneurysm. On average, one of these 54 men will not survive the operation but their aneurysm may never have burst if left untreated'.[29] It may be informative to assess how clear this information is perceived to be compared with or (given that the short length of this text means that there may be scope to develop the explanation further) in addition to information adapted from breast screening. It should also be noted that shortly before recruitment commenced on this study, Public Health England published a revised version of the prostate screening leaflet that included considerably more information on overdiagnosis (and overtreatment).[30] It would be valuable to determine how this new information compares to current information from breast screening (and AAA screening) as it may now be superior.

There was also an association between greater perceived clarity and having previously encountered similar information; the OR was notably high for this variable compared with that of others. To the extent that this represents a causal relationship, this finding suggests that perceived clarity is limited by a general lack of familiarity with the concept. This supports the rationale for ongoing communication campaigns,[1 3–5] in which one of their aims is to create more instances in which people are exposed to information about overdiagnosis. A similar association was observed with previously reading a screening leaflet: although unsurprising, this is reassuring in that it provides some evidence that the leaflets are providing enough background that the concept is perceived as more clear at a later stage (48.6% of participants rated the information extremely/very clear among those who had read a leaflet vs 35.1% of those who had not). This was evident after controlling for exposure to similar information (which was also associated) suggesting that there was a unique effect beyond merely reencountering previously seen information.

Similar to our previous study on engagement with written screening information,[22] there was an association with a 'rational' decision-making style. Participants who were more inclined to make decisions in this way were more likely to find the information extremely or very clear, although it was surprising that this characteristic was associated with perceived clarity, whereas education was not. This finding suggests that the information is not serving individuals as well if they tend not to make decisions in this way. It may be beneficial to explore ways of describing the concept that require less deliberation, such as a more heavily summarised 'gist'[31] or an anecdotal 'narrative' approach,[32 33] describing a personal experience. There was also (weaker) evidence against the

**Table 2** Perceived clarity of overdiagnosis information: descriptive statistics for continuous variables, adjusted ORs, 95% CIs, p values for variables in the multivariable binary logistic regression model

| | | 'How clear do you find this description of a risk of the test?' M (SD) | | | | | Adjusted OR, 95% CI; p-value |
|---|---|---|---|---|---|---|---|
| | **Total** | **Extremely clear** | **Very clear** | **Moderately clear** | **Slightly clear** | **Not at all clear** | **Extremely/very clear** |
| **Characteristic** | **(n=1616)** | **(n=148; 9.2%)** | **(n=542; 33.5%)** | **(n=592; 36.6%)** | **(n=173; 10.7%)** | **(n=161; 10.0%)** | **(vs Not/slightly/ moderately)** |
| Age (in years) | 43.7 (15.7) | 45.4 (14.8) | 44.3 (16) | 43.3 (15.9) | 43.2 (15.7) | 42.4 (14.7) | 1.00, 0.99 to 1.00; 0.282 |
| Decision-making styles | | | | | | | |
| Avoidant score | 13.8 (3.9) | 13.3 (4.4) | 13.8 (3.8) | 13.9 (3.9) | 13.7 (3.9) | 13.5 (3.8) | 1.01, 0.98 to 1.04; 0.713 |
| Dependent score | 16.9 (3.5) | 16.2 (4) | 16.8 (3.5) | 17.0 (3.4) | 17.4 (3.5) | 16.5 (3.4) | 0.97, 0.94 to 1.00; 0.052 |
| Intuitive score | 18.7 (2.8) | 18.8 (3.4) | 18.9 (2.7) | 18.6 (2.7) | 18.7 (2.7) | 18.3 (2.6) | 1.02, 0.98 to 1.06; 0.413 |
| Rational score | 19.7 (2.6) | 20.4 (2.7) | 19.7 (2.6) | 19.6 (2.6) | 19.7 (2.8) | 19.4 (2.6) | 1.06, 1.02 to 1.11; **0.009** |
| Spontaneous score | 14.7 (3.7) | 14.5 (4.1) | 14.9 (3.6) | 14.9 (3.6) | 14.5 (3.8) | 14.0 (3.7) | 1.03, 1.00 to 1.07; 0.073 |

Adjusted ORs and 95% CIs are per unit increase; p values<0.05 are in bold; all predictor variables are included in the model.

null hypothesis for a negative association between dependent decision-making and perceived clarity. This may be mitigatable by providing clear instruction on who an individual can contact for further information (eg, a primary care provider).

This study has limitations. First, generalisability of results may be reduced by the exclusion of participants with incomplete or inapplicable data. Second, although we added some background information regarding the aims and design of screening programmes, information on overdiagnosis was removed from its original context, which includes more detail on practicalities and outcomes of screening. If participants had encountered the information outside of a study context, perceived clarity may have been better (or possibly worse, in the presence of competing information points). The focus on perceived clarity of information meant that we did not measure other important variables such as objective understanding. Future research could quantify the extent to which the descriptions used are perceived as clear in

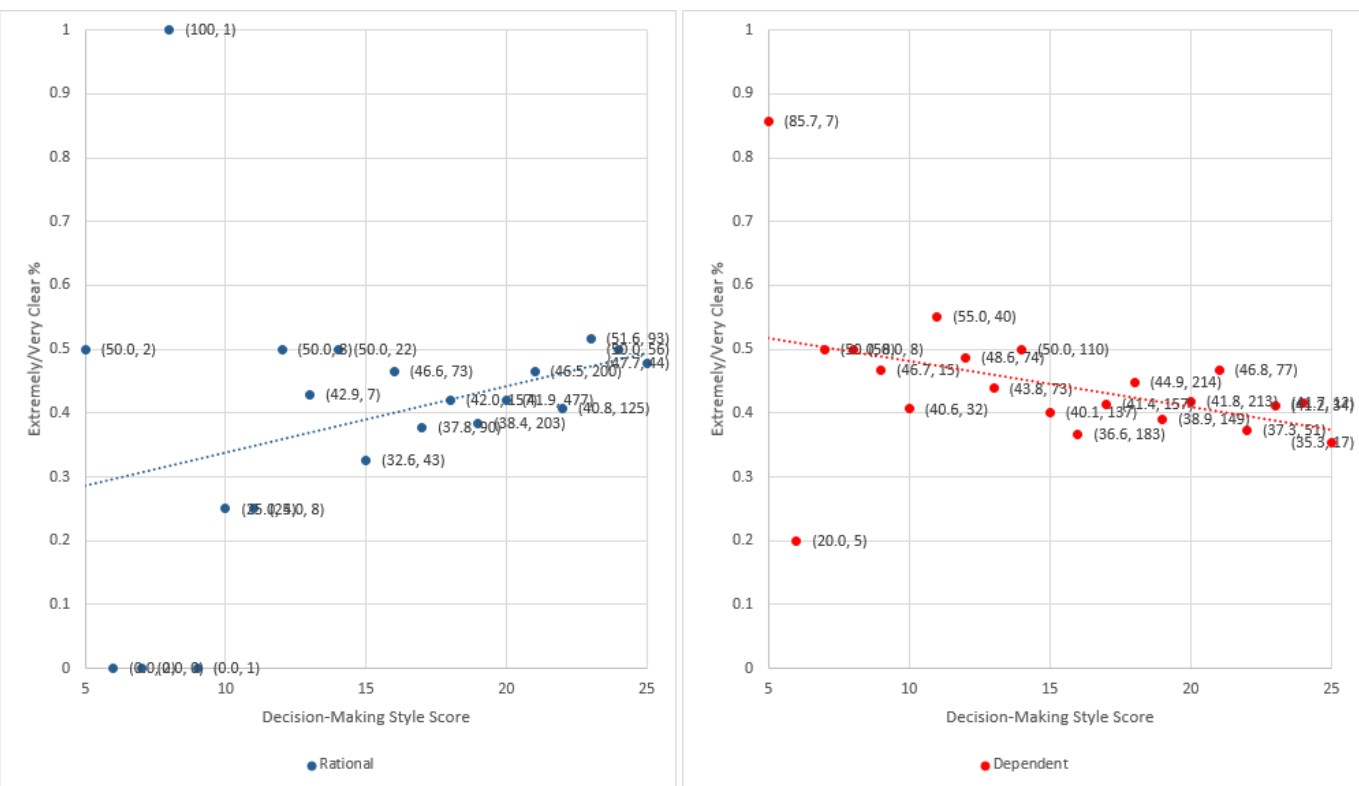

**Figure 1** Proportion of participants who perceived the information as extremely/very clear within each possible score level for rational and dependent decision-making styles (percentage and total number of participants in brackets).

more naturalistic contexts and whether they are understood correctly by more objective standards. Finally, the study aimed to determine predictors of perceived clarity in the general public, meaning that the sample was heterogeneous in terms of eligibility for specific screening modalities and the extent to which they were at risk of overdiagnosis in other healthcare contexts. In areas with a focus on communicating with a more specific group, it may be beneficial to test whether these findings replicate and whether other factors are relevant.

In conclusion, we found that information on overdiagnosis from the Breast Screening Programme in England is more likely to be rated as clearer than equivalent information from the Prostate Cancer Risk Management Programme and may be a valuable template for efforts to communicate the concept. Also, people who were less 'rational' and more 'dependent' in their decision-making may find existing information more challenging, suggesting that other descriptions may be appropriate for these individuals. Future research aiming to improve methods of communicating overdiagnosis information could assess possible descriptions of overdiagnosis in terms of objective knowledge and test for associations within specific groups who may find the concept relevant. These findings provide some evidence that communication can be improved by creating additional opportunities for people to engage with information about the concept (eg, via organised campaigns).

**Contributors** AG, CR and JW conceived and designed the study. AG and EM analysed the data. AG, CR and JW participated in the interpretation. All authors drafted the manuscript, participated in critical revision and approved the final version.

**Funding** The current study was supported by a programme grant from Cancer Research UK awarded to Professor Jane Wardle (C1418/A14134). Dr Jo Waller is supported by a Career Development Fellowship from Cancer Research UK (C7492/A17219). Cancer Research UK was not involved in the design of this study; the collection, analysis or interpretation of the results; in the writing of the manuscript or in the decision to submit for publication.

**Competing interests** None declared.

**Patient consent** Obtained.

**Ethics approval** University College London Research Ethics Committee.

**Provenance and peer review** Not commissioned; externally peer reviewed.

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
