## [Reviewer comments · BMJ Open]

ARTICLE DETAILS

TITLE (PROVISIONAL)	Comparing perceived clarity of information on overdiagnosis used for breast and prostate cancer screening in England: An experimental survey
AUTHORS	Ghanouni, Alex; Renzi, Cristina; McBride, Emily; Waller, Jo

VERSION 1 - REVIEW

REVIEWER	PD Dr. O. Wegwarth Max Planck Institute for Human Development Harding Center for Risk Literacy Lentzeallee 94 14195 Berlin, Germany
REVIEW RETURNED	13-Mar-2017

GENERAL COMMENTS	Review of "Comparing perceived clarity of information on overdiagnosis used for breast cancer and prostate cancer screening in England." The authors seek to shed light on the perceived clarity of information on overdiagnosis that is used for breast cancer screening and prostate cancer screening. I welcome the intention of the paper because it touched upon an often-ignored fact in cancer screening: Screening does not only come with benefits but also with harms. Particularly the harms are rarely known by people who attend screening. To learn more about how people understand currently provided information via screening pamphlets would have been a worthwhile endeavor. Unfortunately, the actual understanding of different descriptions of overdiagnosis was not what the study tested, but rather a qualitative self-rating of whether the descriptions were clear or not. This raises certain issues: 1) Within the abstract and introduction, the paper rightly highlights how important awareness and understanding of overdiagnosis is in the context of screening. It further highlights the limited research into effective methods of communication. What I found irritating in the light of these insights is that the authors eventually ended up with a study on perceived clarity instead of one on actual understanding and testing of ways to improve understanding. 2) Further, it has been established for decades that subjective self-ratings do tell us very little about reality (here, potential understanding). I would have found it helpful if the authors explained in some detail what the readers would eventually learn from the investigated main outcome of "perceived clarity" in the light of such study findings. It should be made clear—particularly in the introduction that highlights the importance of understanding—that "perceived clarity" has an unknown correlation and/or causality with
---

	“actual understanding.” 3) There is a methodological concern with respect to the phrasing of the question “How clear do you find this description of risk...?”—Has any of the authors tested upfront how people understand this question? In my view, it invites a variance of interpretation. While people may find the description of the risk very clear (perceived clarity), they may still not understand the described risk (no actual understanding). 4) The authors introduced the concept of “decision-making styles” and find that people who rate themselves as more “rational” are also those who are more likely to find the description of overdiagnosis clear. Both, “decision-making styles” as well as “clarity rating” are subjective self-evaluations. To play devil’s advocate, I wonder if it goes kind of together that those people who give themselves the “gold standard rating” of good decision making (rational) are, at the same time, also more likely to pretend that there is clarity when it is not necessarily there. Other issues: Analysis, page 9 line 45 to 53 (see also Results and Table 1, page 10, line 45 ff, page 18, line 10 ff): The authors decided to dichotomize the 5-point scale of clarity in a way they considered most conceptually meaningful, which resulted in bringing together “extremely clear” and “very clear” on one side and “moderately clear” to “not at all clear” on the other side. As there is no external criterion, it can be argued back and forth which scale points to combine. From my subjective perspective, however, I find it more intuitive to move “moderately clear” to the side of “extremely clear and very clear” and not to the side of “not clear.” If “moderately clear” would be moved to “extremely clear and very clear” then—following the numbers in Table 1, line 10 ff.—the currently reported difference in clarity between breast and prostate cancer leaflets should be far less strong as it appears right now. Page 4, line 44: When introducing the “decision-making styles,” the authors begin the sentence with “We have previously explored...”—yet, none of the listed authors is the author of the cited work in line 48 (citation no. 20). Participants: The sample included people aged 18 to 70 years. People aged 18 to 50 years are not affected by breast cancer and prostate cancer screening. Why were these included? Does this age group differ from the group at screening age with respect to the clarity rating? The term “social class grading” is somewhat disturbing. I’m sure most readers have never heard of this and the grades A, B, etc. require explanations, although I would suggest to delete this.
--	--

REVIEWER	Dr Ray Moynihan Bond University, Australia
REVIEW RETURNED	18-Apr-2017

GENERAL COMMENTS	I think this is a very interesting and well-designed study about a very important topic, and I only have very minor suggestions for revisions. I have a key interest in overdiagnosis, but importantly, no specialist skills in biostatistics, so I will make no comment on the statistics and recommend someone with biostatistics expertise review the paper. Specific comments I understand that this study was not investigating how clear the
---

materials were, but rather comparing clarity of two different forms of information. However, if there is any chance to be explicit about this in the Introduction or the Method, that would be great. I know you address these general issues in the Discussion, but it currently feels a little odd earlier in the paper not making some mention of the general need for clarity/comprehensibility about this important information.

Related to this comment: it is not clear to me whether the extracted texts provided on Page 6, Lines 7-17, are the totality of the information in leaflets about overdiagnosis, or just short extracts of longer information on overdiagnosis. Forgive me if I have missed something. I think the authors need to be explicit about this – and if this is the totality of the information perhaps some comment needs to be made about this in the Discussion. (eg, on the need for information to be clearer, better, longer, repeated..etc)

Page 5, Lines 28-30, there is a reference to design and “some aspects” of measures described elsewhere (Under review). As a reviewer (and potential reader) this does not feel very helpful, (particularly the use of word “some”, as we have no idea which) and I think you need to say something like...”.... a more detailed explanation of method and measures is available elsewhere...”....and then go on to give some key points about methods and measures.

Page 5, Line 42-43 When you mention the leaflets, it might be good to include them as an extra supporting information file (for those interested)

Page 6, 7 & 8. You might consider putting some of this material into a box, or boxes, along with the options for answers. It feels a little odd having it all in prose. (This is just a suggestion- do not feel you have to do this – as you may have good justification for the current presentation)

Page 9 , Line 6 Consider adding this exclusion criteria to your limitations, as a possible limit of generalisability of findings (as there seem to be a large number excluded)

Page 9 : In the interests of clarity (given the paper may be read by a wide audience with different skills and experiences) I wonder whether you could be a little clearer about some parts of the analysis...– eg Page 9- mention of violation.....and page 10, Lines 11-15, mention of variance inflation factors . Also, I just wanted ask whether the mention of both these issues was best in the Methods or Results- I am unsure, but wanted to double check with authors.

Page 10, Line 44: consider mentioning that these results are in Table 1 – and I suggest using numbers (with CIs etc) for all of the results section.

Page 12, Lines 15-19: re: the extract about AAA – again, its not clear to me if this is all that is said about AAA OD – or whether this is just an extract- please make it clear – and perhaps offer a view on the adequacy of such a small mention of a potentially important issue.

And as a general final comment on the Discussion, consider placing the individual information-provision strategies (eg leaflets) into a

	wider context of multiple strategies to better inform about and address OD (eg possible public awareness campaigns etc). For example the work of Susan Michie argues for a whole suite of activities to better inform and/or change behaviour.
--	--

VERSION 1 – AUTHOR RESPONSE

Reviewer: 1

Reviewer Name: PD Dr. O. Wegwarth

Institution and Country: Max Planck Institute for Human Development, Harding Center for Risk Literacy, Berlin, Germany
Competing Interests: None declared

"The authors seek to shed light on the perceived clarity of information on overdiagnosis that is used for breast cancer screening and prostate cancer screening. I welcome the intention of the paper because it touched upon an often-ignored fact in cancer screening: Screening does not only come with benefits but also with harms. Particularly the harms are rarely known by people who attend screening. To learn more about how people understand currently provided information via screening pamphlets would have been a worthwhile endeavor.

Unfortunately, the actual understanding of different descriptions of overdiagnosis was not what the study tested, but rather a qualitative self-rating of whether the descriptions were clear or not. This raises certain issues:

Within the abstract and introduction, the paper rightly highlights how important awareness and understanding of overdiagnosis is in the context of screening. It further highlights the limited research into effective methods of communication. What I found irritating in the light of these insights is that the authors eventually ended up with a study on perceived clarity instead of one on actual understanding and testing of ways to improve understanding.

Further, it has been established for decades that subjective self-ratings do tell us very little about reality (here, potential understanding). I would have found it helpful if the authors explained in some detail what the readers would eventually learn from the investigated main outcome of "perceived clarity" in the light of such study findings. It should be made clear—particularly in the introduction that highlights the importance of understanding—that "perceived clarity" has an unknown correlation and/or causality with "actual understanding."

AR: We accept the reviewer's assessment that perceived clarity does not necessarily correlate with actual understanding (and based on previous research, we would hypothesise that using self-reported perceived clarity would lead to overestimates of actual understanding if it were used as a proxy measure). We have redrafted sections of the Introduction and Discussion sections to draw a clearer distinction between understanding and perceived clarity. We have argued that both issues are important within the broader context of communication about overdiagnosis, and that perceived clarity has implications for decisional conflict. We also acknowledge in the highlights and the limitations section of the Discussion that objective understanding of information on overdiagnosis is also important to measure and would merit further research.

"There is a methodological concern with respect to the phrasing of the question "How clear do you find this description of risk...?"—Has any of the authors tested upfront how people understand this question? In my view, it invites a variance of interpretation. While people may find the description of the risk very clear (perceived clarity), they may still not understand the described risk (no actual understanding)."

AR: As we note in the Method: Piloting section, measures were tested via cognitive interviews with members of the public. The interview schedule informed participants:

"We want to know whether the wording of the questions is clear, and whether you would be able to answer the questions. You may answer the questions truthfully, but you don't have to. At this stage, we are only interested in whether the questions are clear. If a question is unclear, please let me know."

At the end of the interview, participants were asked up to three questions:

"Were there any items that were difficult to understand, or offensive?"; "What was difficult or offensive about them? Do you have any suggestions for rephrasing these items?"

We recognise that this does not address the previously acknowledged uncertainty in the correlation between perceived clarity and actual understanding. However, it offered participants the opportunity to highlight any issues with wording that occurred to them. Participants were forthcoming with their feedback in general but did not express any major concerns with the question. We have added detail on the cognitive interviews within the Method: Piloting section.

"The authors introduced the concept of "decision-making styles" and find that people who rate themselves as more "rational" are also those who are more likely to find the description of overdiagnosis clear. Both, "decision-making styles" as well as "clarity rating" are subjective self-evaluations. To play devil's advocate, I wonder if it goes kind of together that those people who give themselves the "gold standard rating" of good decision making (rational) are, at the same time, also more likely to pretend that there is clarity when it is not necessarily there."

AR: We appreciate this thoughtful point and agree it is possible that a self-serving bias may have contributed to concurrent higher self-ratings of rationality and perceived clarity. However, we would suggest that rationality is not an inherently 'gold-standard' decision-making style. One could also hypothesise that those who wish to rate themselves as 'quick thinkers' (i.e. more spontaneous decision-makers) may also wish to see themselves as able to rapidly appraise the information and hence would rate it as clearer. Similar hypotheses could be generated for other decision-making styles. Given that we did not observe other potential examples of self-serving bias, we do not think that self-serving perceptions of rationality and self-ratings of perceived clarity "go together" as naturally as might be supposed.

"Other issues: Analysis, page 9 line 45 to 53 (see also Results and Table 1, page 10, line 45 ff, page 18, line 10 ff): The authors decided to dichotomize the 5-point scale of clarity in a way they considered most conceptually meaningful, which resulted in bringing together "extremely clear" and "very clear" on one side and "moderately clear" to "not at all clear" on the other side. As there is no external criterion, it can be argued back and forth which scale points to combine. From my subjective perspective, however, I find it more intuitive to move "moderately clear" to the side of "extremely clear and very clear" and not to the side of "not clear." If "moderately clear" would be moved to "extremely clear and very clear" then—following the numbers in Table 1, line 10 ff.—the currently reported difference in clarity between breast and prostate cancer leaflets should be far less strong as it appears right now."

AR: Although we take the view that dichotomising extremely/very clear vs. less clear is a balanced approach to the analysis, we entirely agree that other readers may regard other thresholds for dichotomisation as more appropriate. In the interests of transparency, we have run sensitivity analyses to generate results from other possible thresholds and reported them in Appendix 1 and

Appendix 2, along with a comment in the Results: Analysis section to explain them to the interested reader. As anticipated, associations are not present for all dichotomisations although some were relatively robust (e.g. having previously read or heard similar information and rational decision-making style).

"Page 4, line 44: When introducing the "decision-making styles," the authors begin the sentence with "We have previously explored..."—yet, none of the listed authors is the author of the cited work in line 48 (citation no. 20)."

AR: We would like to clarify that 20 is a reference to the general concept and measurement of decision-making styles, whereas 21 is the previous study from our group (comprising three of the four authors on this paper). We have amended this paragraph to make this clear.

"Participants: The sample included people aged 18 to 70 years. People aged 18 to 50 years are not affected by breast cancer and prostate cancer screening. Why were these included? Does this age group differ from the group at screening age with respect to the clarity rating?"

AR: We have clarified in the Introduction that we are considering methods of communicating overdiagnosis to the general public (irrespective of a specific context) and are using information from breast and prostate cancer screening leaflets as a template in order to do this.

"The term "social class grading" is somewhat disturbing. I'm sure most readers have never heard of this and the grades A, B, etc. require explanations, although I would suggest to delete this."

AR: We agree that these levels require explanation in order to be meaningful to the reader and have added details and a reference to the Method: Measures: Demographics section. Given that this is a widely used measure of socioeconomic status, we think this is an important variable to include and would limit the validity of our results were we to delete it.

Reviewer: 2

Reviewer Name: Dr Ray Moynihan

Institution and Country: Bond University, Australia Competing Interests: None declared

"I think this is a very interesting and well-designed study about a very important topic, and I only have very minor suggestions for revisions. I have a key interest in overdiagnosis, but importantly, no specialist skills in biostatistics, so I will make no comment on the statistics and recommend someone with biostatistics expertise review the paper.

Specific comments

I understand that this study was not investigating how clear the materials were, but rather comparing clarity of two different forms of information. However, if there is any chance to be explicit about this in the Introduction or the Method, that would be great. I know you address these general issues in the Discussion, but it currently feels a little odd earlier in the paper not making some mention of the general need for clarity/comprehensibility about this important information."

AR: As we outline in our previous response, we have amended the Introduction and Discussion section to distinguish more clearly between objective understanding and perceived clarity.

"Related to this comment: it is not clear to me whether the extracted texts provided on Page 6, Lines 7-17, are the totality of the information in leaflets about overdiagnosis, or just short extracts of longer information on overdiagnosis. Forgive me if I have missed something. I think the authors need to be explicit about this – and if this is the totality of the information perhaps some comment needs to be

made about this in the Discussion. (eg, on the need for information to be clearer, better, longer, repeated..etc)."

AR: We can confirm that this is the totality of information on overdiagnosis contained in the prostate screening information leaflet, as it existed at the time of the study (although please see our comments below regarding the more detailed updated version). The breast screening information leaflet does contain some information that overlaps with the concept of overdiagnosis e.g. on Ductal Carcinoma in Situ:

"About 1 in 5 women diagnosed with breast cancer through screening will have non-invasive cancer. This means there are cancer cells in the breast, but they are only found inside the milk ducts (tubes) and have not spread any further. This is also called ductal carcinoma in situ (DCIS). In some women, the cancer cells stay inside the ducts. But in others they will grow into (invade) the surrounding breast in the future."

And ratios of harms and benefits of breast cancer screening:

"Overall for every 1 woman who has her life saved from breast cancer, about 3 women are diagnosed with a cancer that would never have become life-threatening. Researchers are trying to find better ways to tell which women have breast cancers that will be life-threatening and which women have cancers that will not."

The text used as the basis for the information in the study is also reiterated in other words:

"Some women will be diagnosed and treated for breast cancer that would never otherwise have been found and would not have become life-threatening."

However, we deliberately omitted information that was context-specific (e.g. on DCIS) and focused only on broad conceptual information (hence changing "cancer" to "an illness" in our descriptions). We have acknowledged these points in the Method: Design, where the information on overdiagnosis is described.

"Page 5, Lines 28-30, there is a reference to design and "some aspects" of measures described elsewhere (Under review). As a reviewer (and potential reader) this does not feel very helpful, (particularly the use of word "some", as we have no idea which) and I think you need to say something like..."... a more detailed explanation of method and measures is available elsewhere..."...and then go on to give some key points about methods and measures."

AR: As we acknowledge in our previous response, the intention of this text was to indicate that similar information had already been published, as a guard against the appearance of self-plagiarism. We have amended this text to clarify.

"Page 5, Line 42-43 When you mention the leaflets, it might be good to include them as an extra supporting information file (for those interested)."

AR: We agree that it would be advantageous to append the leaflets for the benefit of the reader. However, we are unclear whether we have implicit permission under copyright law to reproduce them in full and since one is freely available online and the other is available on request, we think this may be unnecessary. We have highlighted that this is the case in the Method: Design section for the benefit of the reader.

"Page 6, 7 & 8. You might consider putting some of this material into a box, or boxes, along with the

options for answers. It feels a little odd having it all in prose. (This is just a suggestion- do not feel you have to do this – as you may have good justification for the current presentation)."

AR: We are grateful for the suggestion. We have considered it and explored various ways of presenting the items in table form but the number of items and response options means that it appears more effective overall to summarise this information in the text.

"Page 9 , Line 6 Consider adding this exclusion criteria to your limitations, as a possible limit of generalisability of findings (as there seem to be a large number excluded)."

AR: We agree that this is a potential limitation and have acknowledged it in the relevant section of the Discussion.

"Page 9 : In the interests of clarity (given the paper may be read by a wide audience with different skills and experiences) I wonder whether you could be a little clearer about some parts of the analysis...– eg Page 9- mention of violation.....and page 10, Lines 11-15, mention of variance inflation factors . Also, I just wanted ask whether the mention of both these issues was best in the Methods or Results- I am unsure, but wanted to double check with authors."

AR: We agree that these issues could be clarified in order to be meaningful to a wider audience and have added explanations of the assumption or proportional odds, Variance Inflation Factors, and the assumption of linearity. Given that these statistics informed the analysis plan and are not results, per se, we believe they are best placed within the Method: Analysis section.

"Page 10, Line 44: consider mentioning that these results are in Table 1 – and I suggest using numbers (with CIs etc) for all of the results section."

AR: We have directed the reader to Table 1 and reiterated the ORs, 95% CIs, and p-values in the Results: Predictors of rating the information on overdiagnosis as extremely/very clear section.

"Page 12, Lines 15-19: re: the extract about AAA – again, its not clear to me if this is all that is said about AAA OD – or whether this is just an extract- please make it clear – and perhaps offer a view on the adequacy of such a small mention of a potentially important issue."

AR: We have clarified that this represents the entirety of information on overdiagnosis in the leaflet and have highlighted how our results might be used in relation to the short length of this text in the Discussion section.

"And as a general final comment on the Discussion, consider placing the individual information-provision strategies (eg leaflets) into a wider context of multiple strategies to better inform about and address OD (eg possible public awareness campaigns etc). For example the work of Susan Michie argues for a whole suite of activities to better inform and/or change behaviour."

AR: We agree that the taxonomy offered by Michie et al., based on the COM-B model, is a valuable guide for determining the types of interventions that may most effectively change behaviour and the contexts in which they are likely to be most effective. However, this study is oriented around a problem that is somewhat different (i.e. a need to improve public communication rather than change any particular behaviour) and so it is not clear that behaviour change theory is directly applicable as a way of taking these findings forward. We have not made additional amendments relating to this point but we would welcome further suggestions, if warranted.

ADDITIONAL COMMENTS FROM THE AUTHORS:

We have become aware that an updated version of the prostate screening leaflet came into effect shortly before our study began recruitment, which means that a new set of information on overdiagnosis exists and could usefully be compared to the information derived from the breast screening leaflet. We have reworded the Discussion section to acknowledge this. However, it has meant that the first implication of our findings no longer applies, as it was previously written.

Please could we also highlight that due to a departmental change at our institution, the authors' affiliations have now been revised from the "Health Behaviour Research Centre, Department of Epidemiology and Public Health" to "Research Department of Behavioural Science and Health".

VERSION 2 – REVIEW

REVIEWER	Dr Ray Moynihan Bond University
REVIEW RETURNED	06-Jun-2017

GENERAL COMMENTS	Generally, I think the piece is improved, and is ready for publication. I have one final comment- which has only occurred to me, on this second re-revision reading. It strikes me that the manuscript seems to play down this important finding –(and give more prominence to other findings): "Participants were more likely to perceive the information as more clear if they had previously encountered similar information (OR: 1.77, 1.40 to 2.23;p<.0005)" It seems to me, as someone who works a lot in the field of overdiagnosis, that this may be a very important finding – because it may be that one of the ways to address the complex and counter-intuitive nature of overdiagnosis information/explanation, is to try and facilitate multiple opportunities for people to hear and learn about it. I think this paper could make much more of this finding- and potentially give it equal or more prominence than other findings (which seem to have much less impressive ORs) I suggest putting this finding into the conclusion of the abstract – and in the Discussion, giving it equal (at least) prominence with the other findings (ie exposure to previous screening info, and the decision making style) - both on Page 12- and in the concluding remarks at the bottom of page 14. Thanks for the opportunity to review, and thanks to the authors for this work. (and I ticked the N/A button re stats, given my lack of stats training)
---

VERSION 2 – AUTHOR RESPONSE

We appreciate the helpful suggestion from Reviewer 2 and agree that this is a pertinent finding that had not previously been given due prominence. As suggested, we have added comments to the Discussion section (including the Conclusion) and the Abstract.